# Query Optimization for Ontology-Mediated Query Answering

## ABSTRACT

Ontology-mediated query answering (OMQA) consists in asking database queries on knowledge bases (KBs); a KB is a set of facts called the KB's database, which is described by domain knowledge called the KB's ontology. A widely-investigated OMQA technique is FO-rewriting: every query asked on a KB is reformulated w.r.t. the KB's ontology, so that its answers are computed by the relational evaluation of the query reformulation on the KB's database. Crucially, because FO-rewriting compiles the domain knowledge relevant to queries into their reformulations, query reformulations may be complex and their optimization is the crux of efficiency.

We devise a novel optimization framework for a large set of OMQA settings that enjoy FO-rewriting: conjunctive queries, i.e., the core select-project-join queries, asked on KBs expressed in datalog± and existential rules, description logic and OWL, or RDF/S. We optimize the query reformulations produced by state-of-the-art FO-rewriting algorithms by computing rapidly, with the help of a KB's database summary, simpler (contained) queries with same answers that can be evaluated faster by RDBMSs. We show on a well-established OMQA benchmark that time performance is significantly improved by our optimization framework in general, up to three orders of magnitude.

## CCS CONCEPTS

• **Information systems** → **Query optimization**; *Semantic web description languages*; • **Computing methodologies** → *Knowledge representation and reasoning*.

## KEYWORDS

Existential rules, query optimization, data summarization

**ACM Reference Format:**
Anonymous Author(s). 2024. Query Optimization for Ontology-Mediated Query Answering. In *Proceedings of ACM Web Conference 2024 (WWW '24)*. ACM, New York, NY, USA, 10 pages. https://doi.org/XXXXXXX.XXXXXXX

## 1 INTRODUCTION

Ontology-mediated query answering [9] (OMQA) is a widely studied data management problem in Artificial Intelligence, Databases and Semantic Web. It consists in asking database-style queries on knowledge bases (KBs). A KB is a first-order (FO) theory that consists of a set of facts called a database, which models the application's data, and of a set of axioms called an ontology, which models the application's domain knowledge. The notable difference

| Language | First-Order Logic syntax | Relational algebra syntax |
|---|---|---|
| CQ | $q(\bar{x}) = \exists \bar{y} \bigwedge_{i=1}^{n} atom_i$ | $q(\bar{x}) = \Pi_{\bar{x}}(\bowtie_{i=1}^{n} atom_i)$ |
| UCQ | $q(\bar{x}) = \bigvee_{i=1}^{n} CQ_i$ | $q(\bar{x}) = \bigcup_{i=1}^{n} CQ_i$ |
| JUCQ | $q(\bar{x}) = \bigwedge_{i=1}^{n} UCQ_i$ | $q(\bar{x}) = \Pi_{\bar{x}}(\bowtie_{i=1}^{n} UCQ_i)$ |
| SCQ | $q(\bar{x}) = \exists \bar{y} \bigwedge_{i=1}^{n} \bigvee_{j=1}^{m_i} atom_i^j$ | $q(\bar{x}) = \Pi_{\bar{x}}(\bowtie_{i=1}^{n} \bigcup_{j=1}^{m_i} atom_i^j)$ |
| USCQ | $q(\bar{x}) = \bigvee_{i=1}^{n} SCQ_i$ | $q(\bar{x}) = \bigcup_{i=1}^{n} SCQ_i$ |

**Table 1: Main FO query languages used for FO-rewriting**

| KB language | Query reformulation language | | | |
|---|---|---|---|---|
| | UCQ | USCQ | JUCQ | datalog$^{nr}$ |
| datalog±/existential rules | [31, 32, 38] | [51] | | [33, 45] |
| description logics/OWL | [17, 21, 46, 52] | | [12] | [47] |
| RDF/S | [10, 30] | | [11] | |

**Table 2: Main related works on conjunctive query answering via FO-rewriting**

between this query answering setting and the traditional database one is that the answers to queries must be computed w.r.t. both the facts that are stored in the KB's database and the facts that can be deduced from the KB's database with the help of the KB's ontology.

There exist two main OMQA techniques in the literature. Both reduce OMQA to standard query evaluation on relational databases. The first technique is called FO-rewriting, e.g., [17]. It consists in rewriting a query asked on a KB into a so-called query reformulation, so that the query answers are obtained by evaluating the query reformulation on the KB's database. The second technique is called materialization, e.g., [1]. It consists in adding to the KB's database all the facts than can be deduced from it with KB's ontology, so that the query answers are obtained by evaluating the (original) query on the augmented KB's database. The combination of FO-rewriting and materialization, called the combined or hybrid approach, has also been investigated, e.g., [40]. Crucially, both FO-rewriting and materialization are useful because, although there exist simple OMQA settings in which they compete, e.g., [3], there also exist more expressive OMQA settings to which only a single technique applies, e.g., [8].

In this paper, we focus on FO-rewriting, which was introduced in [17]. This technique has been largely studied in OMQA settings consisting of (e.g., Table 2): queries expressed as conjunctive queries (CQs); KBs expressed using datalog± and existential rules, description logics and OWL, or RDF/S; query reformulations expressed as unions of CQs (UCQs) and non-recursive datalog programs (datalog$^{nr}$) that unfold to UCQs, unions of semi-CQs (USCQs), or joins of UCQs (JUCQs). These languages are recalled in Table 1. We consider all these OMQA settings in this work.

Standard OMQA via FO-rewriting is illustrated in Figure 1. It consists in producing a query reformulation $q^O$ from a query $q$ and the ontology $O$ of the KB $\mathcal{K}$, and then in evaluating $q^O$ on the database $\mathcal{D}$ of $\mathcal{K}$ stored in an RDBMS. We point out that a query reformulation $q^O$ may be large and complex to evaluate, e.g., [12, 33, 51]. FO-rewriting is indeed both ontology-dependent and data-independent, hence $q^O$ must accomodate to all the possible databases and cannot be specific to the particular database $\mathcal{D}$ of $\mathcal{K}$.

**Figure 1: Standard (solid and dashed, back edges) and optimized (black and blue, solid edges) OMQA via FO-rewriting**

So far, and similarly to semantic query optimization for deductive databases, e.g., [19], query optimization for FO-rewriting has focused on studying equivalent representations of query reformulations that can be evaluated faster: minimal (e.g., [21, 38]), compact (e.g., [33, 51]) or cost-based (e.g., [11, 12]) reformulations. However, because these optimizations are ontology-dependent and data-independent, optimized query reformulations remain complex to evaluate. They correspond to syntactically different but semantically equivalent variants of non-optimized query reformulations, thus they still need to accommodate to all the possible databases, and not to just the fixed database at hand.

The main contribution of this paper is a novel optimization framework for OMQA via FO-rewriting. It is illustrated in Figure 1. This framework capitalizes on the ontology-dependent and data-independent query optimization for FO-rewriting that have been studied so far in the literature (Reformulation step in Figure 1). Its originality is to include complementary data-dependent query optimization for FO-rewriting (Summarization and Optimization steps in Figure 1). Its purpose is to optimize the query reformulation $q^O$ produced by any off-the-shelf FO-rewriting algorithm into a query reformulation $q^{\mathcal{K}}$ that is optimized for the particular database $\mathcal{D}$ of $\mathcal{K}$: $q^{\mathcal{K}}$ is simpler than $q^O$, as it just needs to accommodate to $\mathcal{D}$, so that it can be evaluated faster; at the same time it has the same answers as $q^O$ on $\mathcal{D}$ in order to guarantee the correctness of query answering on $\mathcal{K}$. Crucially, $q^O$ is optimized for $\mathcal{D}$ using a summary $\mathcal{S}$ of $\mathcal{D}$, which is a typically small approximation of $\mathcal{D}$. This allows a trade-off between optimization time and the extent to which $q^{\mathcal{K}}$ is optimized for $\mathcal{D}$.

More specifically, our optimization framework builds on the following contributions.

1/ We formalize the problem of data-dependent optimization of a query reformulation using the well-known notion of query containment [1] (Section 3).

2/ We devise an optimization function $\Omega$ that rewrites a query reformulation into a simpler contained one, i.e., a simpler more specific one, with the same answers on a fixed database (Section 4.1). Containment and query answering correctness are ensured by appropriately removing useless subqueries from the query reformulation, i.e., subqueries that do not participate in producing answers on the given database while they may take time to be evaluated.

3/ We define a summary of a database, which is a (typically much smaller) homomorphic database (Section 4.1). A summary can be used by our $\Omega$ optimization function in place of the original database to perform faster a sound but incomplete identification and removal of useless subqueries (i.e., some useless subqueries may not be removed with a summary). Then, we adapt the quotient operation from graph theory [34] in order to build concrete summaries

tailored to our needs (Section 4.2): both small summaries for fast optimization time and precise summaries to limit the incompleteness of identifying useless subqueries.

4/ We experimentally evaluate our optimization framework on the well-established LUBM$^{\exists}$ benchmark for DL-lite$_{\mathcal{R}}$ KBs (Section 5). DL-lite$_{\mathcal{R}}$ is the description logic that underpins the W3C's OWL2 QL profile for OMQA on large KBs [17]. We show that our optimization framework significantly improves query answering time performance (up to 3 orders of magnitude).

The paper is organized as follows. We present OMQA and FO-rewriting in Section 2. We introduce our optimization framework and we formalize the underlying research problem in Section 3. We devise a solution ($\Omega$ optimization function and database summaries) to this problem in Section 4 and we experimentally evaluate this solution in Section 5. Finally, we conclude with related work and perspectives in Section 6. Proofs are available in the appendix.

This paper is an in-depth presentation of our optimization framework that was briefly introduced in the short paper [5].

## 2 PRELIMINARIES

**KBs.** We consider FO KBs expressed using datalog± or existential rules [8, 13–16], which we simply call *rules* hereafter. A KB $\mathcal{K}$ is of the form $\mathcal{K} = (O, \mathcal{D})$, where $O$ is the KB's *ontology* and $\mathcal{D}$ is the KB's *database*. An ontology $O$ is a set of rules of the form $\forall \bar{x}(q_1(\bar{x}) \rightarrow q_2(\bar{x}))$, where $q_1$ and $q_2$ are CQs (recall Table 1) with the same set $\bar{x}$ of answer variables. Rules are used to derive entailed facts in the KB. A database $\mathcal{D}$ is a set of *incomplete facts*, i.e., whose *terms* are constants and existential variables modeling unknown values [1, 37], which we simply call *facts* from now. The semantics of a KB $\mathcal{K} = (O, \mathcal{D})$ is that of the FO formula $\bigwedge_{\text{rule} \in O} \wedge \exists \bar{v}(\bigwedge_{\text{fact} \in \mathcal{D}})$, where $\bar{v}$ is the set of variables that appear in $\mathcal{D}$.

*Notation.* We use small letters to denote constants, e.g., f, h, etc., and small italic letters to denote variables, e.g., $x$, $y$, etc. Also, we omit quantifiers in rules: existential variables are those that solely appear on the right-hand side of $\rightarrow$ by virtue of FO semantics[1].

*Example 2.1 (Running example).* Let us consider the following DL-lite$_{\mathcal{R}}$ KB $\mathcal{K} = (O, \mathcal{D})$, here expressed using rules [14]:
$O = \{r_1 = ww(x, y) \rightarrow ww(y, x), r_2 = sup(x, y) \rightarrow ww(x, y),$
$\quad r_3 = PhD(x) \rightarrow sup(y, x)\},$
$\mathcal{D} = \{R(\mathsf{f}), R(\mathsf{h}), sup(\mathsf{f}, \mathsf{w}), sup(\mathsf{h}, \mathsf{w}), PhD(\mathsf{w}), ww(\mathsf{f}, \mathsf{h}), R(\mathsf{u}),$
$\quad ww(\mathsf{u}, \mathsf{c}), PhD(\mathsf{c})\}.$
The ontology $O$ states that working with ($ww$) someone is a symmetric relation ($r_1$), supervising someone ($sup$) is a particular case of working with someone ($r_2$), and PhD students ($PhD$) are necessarily supervised ($r_3$). The database $\mathcal{D}$ states that f and h are researchers ($R$) who supervise the PhD student w, f works with h, and a researcher $u$ works with the PhD student c. ◇

**Ontology-mediated query answering.** We consider *FO queries* of the form $q(\bar{x}) = \phi$, where $\phi$ is an FO formula, the set of free (non-quantified) variables of which is exactly the tuple $\bar{x}$ of answer variables. The arity of a query $q(\bar{x})$ is the cardinality of $\bar{x}$; $q(\bar{x})$ is said *Boolean* if $\bar{x} = \emptyset$. A *certain answer* to a query $q(\bar{x})$ of arity $n$ on a KB $\mathcal{K}$ is a tuple $\bar{\mathsf{t}}$ of $n$ constants from $\mathcal{K}$ such that $\mathcal{K} \models q(\bar{\mathsf{t}})$,

---

[1] $\forall \bar{x}(\exists \bar{y} \wedge_{i=1}^{m} a_i \rightarrow \exists \bar{z} \wedge_{j=1}^{n} b_j) \Leftrightarrow \forall \bar{x}(\neg(\exists \bar{y} \wedge_{i=1}^{m} a_i) \vee \exists \bar{z} \wedge_{j=1}^{n} b_j) \Leftrightarrow$
$\forall \bar{x}(\forall \bar{y} \neg(\wedge_{i=1}^{m} a_i) \vee \exists \bar{z} \wedge_{j=1}^{n} b_j) \Leftrightarrow \forall \bar{x} \forall \bar{y}(\wedge_{i=1}^{m} a_i \rightarrow \exists \bar{z} \wedge_{j=1}^{n} b_j)$

where $q(\bar{\mathsf{t}})$ is the Boolean query obtained by instantiating $\bar{x}$ with $\bar{\mathsf{t}}$ in $q$; when $q$ is Boolean, $\bar{\mathsf{t}}$ is the empty tuple $\langle\rangle$. From now, we denote by $ans(q, \mathcal{K})$ the *answer set* of $q$ on $\mathcal{K}$ and we remark that if $q$ is Boolean then the answer is true when $ans(q, \mathcal{K}) = \{\langle\rangle\}$ and the answer is false when $ans(q, \mathcal{K}) = \emptyset$.

*Example 2.2 (Cont.).* Let us consider the CQ (recall Table 1) asking for the supervisees who work with h that must be a researcher: $q(x) = \exists y\, R(\mathsf{h}) \wedge ww(\mathsf{h}, x) \wedge sup(y, x)$.

Its answer set on $\mathcal{K}$ is $ans(q, \mathcal{K}) = \{\mathsf{w}\}$: w is obtained from $R(\mathsf{h}) \in \mathcal{D}$, $sup(\mathsf{f}, \mathsf{w}) \in \mathcal{D}$ or $sup(\mathsf{h}, \mathsf{w}) \in \mathcal{D}$, and the fact $ww(\mathsf{h}, \mathsf{w})$ entailed from $sup(\mathsf{h}, \mathsf{w}) \in \mathcal{D}$ and $r_2$. ◇

**Ontology-mediated query answering technique.** We focus on optimizing OMQA via *FO-rewriting* [17].

FO-rewriting reduces query answering on KBs to query evaluation on relational databases in *FO-rewritable OMQA settings*. An OMQA setting is a pair $(\mathcal{L}_Q, \mathcal{L}_K)$ of query and KB languages. Such a setting is FO-rewritable if for any $\mathcal{L}_Q$ query $q$ and any $\mathcal{L}_K$ ontology $O$, there exists an FO query $q^O$, called a *reformulation of $q$ w.r.t. $O$*, such that for any KB $\mathcal{K} = (O, \mathcal{D})$: $ans(q, \mathcal{K}) = eval(q^O, \mathcal{D})$, where $eval(q^O, \mathcal{D})$ the relational evaluation of $q^O$ on $\mathcal{D}$. Furthermore, each FO-rewriting algorithm computes query reformulations in a fixed FO query dialect. Recall for instance Table 2 where CQs are reformulated into UCQs, USCQs, JUCQs or datalog$^{nr}$ programs. We therefore term *FO-rewriting setting* a triple of query language $\mathcal{L}_Q$, KB language $\mathcal{L}_K$ and query reformulation language $\mathcal{L}_R$, denoted by $(\mathcal{L}_Q, \mathcal{L}_K, \mathcal{L}_R)$, such that $(\mathcal{L}_Q, \mathcal{L}_K)$ is an FO-rewritable OMQA setting for which query reformulations are expressed in $\mathcal{L}_R$.

In this paper, we focus on FO-rewriting settings with queries expressed in the language of CQs and query reformulations expressed in the languages of UCQs, USCQs and JUCQs. These setting are widely considered in the literature on FO-rewriting (e.g., Table 2). We remark that datalog$^{nr}$ reformulations must be unfolded into UCQs reformulations, which we consider, to be evaluated by RDBMSs.

A key property of the FO-rewriting settings that we consider, on which our work relies, is that a query reformulation $q^O$ is equivalent to the CQ $q$ w.r.t. $O$. In particular, $q^O$ is equivalent, regardless of the language used to express it, to the union of all the *CQs that are maximally-contained in $q$ w.r.t. $O$*, i.e., the union of all the most general CQ specializations of $q$ w.r.t. $O$. We recall that (*i*) a query $q'$ is *contained in a query $q$*, denoted by $q' \subseteq q$, if and only if for each database $\mathcal{D}$, $eval(q', \mathcal{D}) \subseteq eval(q, \mathcal{D})$ and (*ii*) a query $q'$ is *contained in a query $q$ w.r.t. an ontology $O$*, denoted by $q' \subseteq_O q$, if and only if for each KB $\mathcal{K} = (O, \mathcal{D})$, $ans(q', \mathcal{K}) \subseteq ans(q, \mathcal{K})$. A query $q'$ is *maximally-contained in a query $q$ w.r.t. an ontology $O$* if and only if (*i*) $q' \subseteq_O q$ and (*ii*) for any other query $q'' \subseteq_O q$, if $q' \subseteq q''$ then $q'' \subseteq q'$ (i.e., $q'$ and $q''$ are equivalent).

*Notation.* We omit existential quantifiers in queries, as non-answer variables are existentially quantified in the query languages we consider (recall Table 1). For instance, the CQ of Example 2.2 is now written $q(x) = R(\mathsf{h}) \wedge ww(\mathsf{h}, x) \wedge sup(y, x)$.

*Example 2.3 (Cont.).* Consider the following equivalent UCQ $q^{\text{UCQ}}$, USCQ $q^{\text{USCQ}}$ and JUCQ $q^{\text{JUCQ}}$ reformulations of $q$ w.r.t. $O$, which are respectively computed by the Rapid [21], Compact [51] and GDL [12] FO-rewriting tools:

$$
\begin{aligned}
q^{\text{UCQ}}(x) = &(R(\mathsf{h}) \wedge ww(\mathsf{h}, x) \wedge sup(y, x)) && (1) \\
&\vee (R(\mathsf{h}) \wedge ww(\mathsf{h}, x) \wedge PhD(x)) && (2) \\
&\vee (R(\mathsf{h}) \wedge sup(\mathsf{h}, x)) && (3) \\
&\vee (R(\mathsf{h}) \wedge ww(x, \mathsf{h}) \wedge sup(y, x)) && (4) \\
&\vee (R(\mathsf{h}) \wedge ww(x, \mathsf{h}) \wedge PhD(x)) && (5) \\
&\vee (R(\mathsf{h}) \wedge sup(x, \mathsf{h}) \wedge sup(y, x)) && (6) \\
&\vee (R(\mathsf{h}) \wedge sup(x, \mathsf{h}) \wedge PhD(x)) && (7)
\end{aligned}
$$

$$
\begin{aligned}
q^{\text{USCQ}}(x) = &(R(\mathsf{h})) \\
&\wedge (ww(\mathsf{h}, x) \vee sup(\mathsf{h}, x) \vee ww(x, \mathsf{h}) \vee sup(x, \mathsf{h})) \\
&\wedge (sup(y, x) \vee PhD(x))
\end{aligned}
$$

$$
\begin{aligned}
q^{\text{JUCQ}}(x) = &(R(\mathsf{h})) \wedge ((ww(\mathsf{h}, x) \wedge sup(y, x)) \\
&\vee (ww(\mathsf{h}, x) \wedge PhD(x)) \\
&\vee (sup(\mathsf{h}, x)) \\
&\vee (ww(x, \mathsf{h}) \wedge sup(y, x)) \\
&\vee (ww(x, \mathsf{h}) \wedge PhD(x)) \\
&\vee (sup(x, \mathsf{h}) \wedge sup(y, x)) \\
&\vee (sup(x, \mathsf{h}) \wedge PhD(x)))
\end{aligned}
$$

$q^{\text{UCQ}}$ is the union of all the maximally-contained CQs in $q$ w.r.t. $O$. $q^{\text{USCQ}}$ and $q^{\text{JUCQ}}$ model the same union up to the distributive property of $\wedge$ and $\vee$. The answer to $q$ on $\mathcal{K}$ (i.e., w) results from (3) in $q^{\text{UCQ}}$, shown in blue, and from the logical combination of the subqueries shown in blue in $q^{\text{USCQ}}$ and $q^{\text{JUCQ}}$, from which (3) can be recovered by distributing the $\wedge$'s over the $\vee$'s. ◇

# 3 OPTIMIZATION PROBLEM

**Motivation.** The definition of FO-rewriting is data-independent: a single query reformulation $q^O$ is able to answer the CQ $q$ on all the KBs with ontology $O$. This generality of $q^O$ follows from the fact that it is equivalent to the union of all the CQs that are maximally-contained in $q$ w.r.t. $O$, which can also be regarded as all the ways databases may store answers to $q$ according to $O$. As a consequence, a query reformulation may be large and complex to evaluate in practice [11, 12, 51]. For instance, the worst-case number of CQs that are maximally-contained in a CQ $q$ w.r.t. a lightweight RDFS, DL-lite$_\mathcal{R}$ or datalog$\pm_0$ ontology, is exponential in the size of the CQ $q$ (number of atoms) [10, 17, 30, 32].

**Rationale behind our optimization problem.** We study the data-dependent optimization of a query reformulation for a particular KB, in order to trade its generality for more OMQA performance. When the query $q$ is asked on a given KB $\mathcal{K} = (O, \mathcal{D})$, the database $\mathcal{D}$ is indeed fixed and just one of all the possible databases a reformulation $q^O$ accommodates to. In particular, within the union of maximally-contained CQs to which $q^O$ is equivalent, many CQs may be *irrelevant* to $\mathcal{D}$, because they have no answer on $\mathcal{D}$ (i.e., $\mathcal{D}$ do not store answers to $q$ w.r.t. $O$ this way), and translate into wasteful evaluation time.

*Example 3.1 (Cont.).* In $q^{\text{UCQ}}$, all the CQs except the CQ (3) are irrelevant to $\mathcal{D}$, and similarly in $q^{\text{USCQ}}$ and $q^{\text{JUCQ}}$, where these CQs are present up to the distribution of the $\wedge$'s over the $\vee$'s. ◇

**Problem statement.** Our goal is to devise an optimization framework for OMQA via FO-rewriting that enjoys the following properties: *generality* to be used in as many FO-rewriting settings as possible, *correctness* to compute the exact answer set of a query, and *effectiveness* to improve query answering time performance.

Our framework relies on an optimization function $\Omega$ that turns a given query reformulation $q^O$ into an *optimized query reformulation*

for a given database $\mathcal{D}$. This optimized query reformulation is hereafter denoted by $q^{\mathcal{K}}$ as it is specific to the KB $\mathcal{K} = (O, \mathcal{D})$.

For the generality of our framework, the $\Omega$ function optimizes query reformulations from the language of $(\wedge, \vee)$-*combinations of CQs* (Definition 3.2 below). Our framework thus applies to FO-rewriting settings with reformulation languages in $(\wedge, \vee)$-combinations of CQs, e.g., UCQ, USCQ and JUCQ.

*Definition 3.2 ($(\wedge, \vee)$-combination of CQs).* A $(\wedge, \vee)$-*combination of CQs*, denoted by $(\wedge, \vee)$-CQ, is either a CQ or a conjunction or union of $(\wedge, \vee)$-CQs.

The $\Omega$ function computes an optimized query reformulation $q^{\mathcal{K}}$ contained in $q^O$ (item 1 in Problem 1 below) since $q^O$ is equivalent to a union of maximally-contained queries, in which we remove those irrelevant to a database $\mathcal{D}$, and removing disjuncts from a union makes it more specific. However, this containment relationship only ensures that the answers to $q^{\mathcal{K}}$ form a subset of the answers to $q^O$ on all the possible databases. For the correctness of our framework, $\Omega$ thus computes an optimized query reformulation $q^{\mathcal{K}}$ with same answers as $q^O$ on $\mathcal{D}$ (item 2 in Problem 1 below).

Finally, for the effectiveness of our framework, the $\Omega$ function optimizes $q^O$ for $\mathcal{D}$ using a summary $\mathcal{S}$ of $\mathcal{D}$ (item 3 in Problem 1 below). This allows a trade-off between the number of removed irrelevant maximally-contained queries and $\Omega$'s runtime, i.e., optimization time. As we shall see in our experiments, the optimization time may be too high to improve OMQA time performance when $\Omega$ identifies irrelevant maximally-contained queries in $q^O$ with the database $\mathcal{D}$ instead of a typically much smaller summary $\mathcal{S}$ of it.

We summarize the above discussion with the formal statement of the research problem studied in this paper.

PROBLEM 1 (OPTIMIZATION FOR OMQA VIA FO-REWRITING). *Let $q^O$ be a $(\wedge, \vee)$-CQ query reformulation and let $\mathcal{D}$ be a database. Define an optimization function $\Omega$ and a summary $\mathcal{S}$ of $\mathcal{D}$ so that the optimization of $q^O$ for $\mathcal{D}$ using $\mathcal{S}$, denoted by $q^{\mathcal{K}}$ and computed by $\Omega(q^O, \mathcal{S})$, satisfies:*

(1) $q^{\mathcal{K}} \subseteq q^O$,
(2) $eval(q^{\mathcal{K}}, \mathcal{D}) = eval(q^O, \mathcal{D})$,
(3) $\tau(\Omega(q^O, \mathcal{S})) + \tau(eval(q^{\mathcal{K}}, \mathcal{D})) \leq \tau(eval(q^O, \mathcal{D}))$, *with $\tau(\cdot)$ the time to compute $\cdot$ in a fixed experimental setup.*

We remark that, above, item 1 cannot be safely removed from Problem 1 since, with only items 2 and 3, $q^{\mathcal{K}}$ may be an arbitrary query with the same answer(s) as $q^O$. E.g., "Where does The Web Conference 2024 take place?" may be optimized by "Where does Petra live?" just because their same answer is Singapore.

## 4 OPTIMIZATION FRAMEWORK

### 4.1 The $\Omega$ optimization function

**Rationale behind the $\Omega$ optimization function.** When a query reformulation is seen as a $(\wedge, \vee)$-combination of CQs, these subCQs are parts of the maximally-contained CQs that the query reformulation models. Recall for instance Example 2.3 where the maximally-contained CQ (3) in the UCQ reformulation corresponds to the logical combinations of the subCQs shown in blue in the JUCQ and USCQ reformulations. Removing subCQs from a query reformulation seen as $(\wedge, \vee)$-combinations of CQs obviously removes all the

maximally-contained queries these subCQs are part of, and crucially for us, removing such subCQs with no answer on a particular database removes maximally-contained queries that are irrelevant to this database. E.g., in Example 2.3, removing from $q^{\text{USCQ}}$ the subCQ $sup(x, \mathsf{h})$ with no answer on $\mathcal{D}$ also removes from $q^{\text{USCQ}}$ the two irrelevant maximally-contained CQs $R(\mathsf{h}) \wedge sup(x, \mathsf{h}) \wedge sup(x, y)$ and $R(\mathsf{h}) \wedge sup(x, \mathsf{h}) \wedge PhD(x)$: without $sup(x, \mathsf{h})$, they cannot be recovered by distributing the $\wedge$'s over the $\vee$'s. We therefore devise the $\Omega$ function to optimize a $(\wedge, \vee)$-CQ query reformulation for a given database by rewriting it from the bottom up to (*i*) identify subCQs with no answer on this database and (*ii*) propagate the effect of their removal within the query reformulation.

**Identifying CQs with no answer on a database.** Checking if a single CQ has no answer on a database can be done easily (e.g., using EXISTS in SQL) and efficiently in general since RDBMSs are highly-optimized for CQs, e.g., [48]. However, doing the same check for all the subCQs in a query reformulation may take significant time, especially when the database is large. To mitigate this issue, $\Omega$ uses database summaries that are (typically small) homomorphic approximations of the databases they summarize. Using such summaries instead of the databases allows trading completeness of identifying subCQs with no answer for efficiency, while retaining soundness.

*Definition 4.1 (Summary of a database).* A database $\mathcal{S}$ is a *summary* of a database $\mathcal{D}$ iff (*i*) there exists a homomorphism $\sigma$ from $\mathcal{D}$ to $\mathcal{S}$, i.e., $\mathcal{D}_\sigma = \mathcal{S}$ where $\mathcal{D}_\sigma$ is the database obtained from $\mathcal{D}$ by replacing the terms[2] in $\mathcal{D}$ by their images in $\mathcal{S}$ through $\sigma$, such that (*ii*) $\sigma$ maps constants in $\mathcal{D}$ to constants in $\mathcal{S}$, while it maps variables in $\mathcal{D}$ to constants or variables in $\mathcal{S}$.

In the above definition, (*i*) ensures that $\mathcal{S}$ is a homomorphic approximation of $\mathcal{D}$, while (*ii*) ensures the soundness of identifying CQs with no answer on $\mathcal{D}$ using $\mathcal{S}$ (Theorem 4.2 below). Also, we remark that a database is a particular summary of itself: $\mathcal{D} = \mathcal{S}$ holds when the database-to-summary homomorphism $\sigma$ maps each term to itself, i.e., when $\sigma$ is the identity function.

THEOREM 4.2. *Let $\mathcal{D}$ be a database and $\mathcal{S}$ a summary of it with the homomorphism $\sigma$. Let $q$ be a CQ $q$ asked on $\mathcal{D}$ and $q_\sigma$ the CQ obtained from $q$ by replacing its constants with their images through $\sigma$. If $q_\sigma$ has no answer on $\mathcal{S}$, then $q$ has no answer on $\mathcal{D}$.*

We stress that, as exemplified below, if $q_\sigma$ has no answer on $\mathcal{S}$ then for sure $q$ has no answer on $\mathcal{D}$, while if $q_\sigma$ has some answer on $\mathcal{S}$ then $q$ may or may not have an answer on $\mathcal{D}$.

*Example 4.3 (Cont.).* Consider the summary $\mathcal{S}$ of $\mathcal{D}$ with homomorphism $\sigma$ such that $\sigma(\mathsf{c}) = \sigma(\mathsf{w}) = \mathsf{p}$, $\sigma(\mathsf{f}) = \sigma(\mathsf{h}) = \sigma(\mathsf{u}) = \mathsf{r}$:

$$\mathcal{S} = \{R(\mathsf{r}), sup(\mathsf{r}, \mathsf{p}), PhD(\mathsf{p}), ww(\mathsf{r}, \mathsf{r}), ww(\mathsf{r}, \mathsf{p})\}.$$

Consider the CQs (1) and (5) in $q^{\text{UCQ}}$, which we name $q^1$ and $q^5$ respectively. By Theorem 4.2: $q_\sigma^1(x) = R(\mathsf{r}) \wedge ww(\mathsf{r}, x) \wedge sup(y, x)$ has an answer on the summary $\mathcal{S}$ ($ans(q_\sigma^1, \mathcal{S}) = \{\mathsf{p}\}$) then $q^1$ may or may not have an answer on $\mathcal{D}$ (here, $q^1$ has no answer on $\mathcal{D}$), while $q_\sigma^5(x) = R(\mathsf{r}) \wedge ww(x, \mathsf{r}) \wedge PhD(x)$ has no answer on $\mathcal{S}$ then for sure $q^5$ has no answer on $\mathcal{D}$.                                                   ◇

$(\wedge, \vee)$-**CQ optimization for a database.** Our $\Omega$ function builds on Theorem 4.2 to optimize a $(\wedge, \vee)$-CQ for a database $\mathcal{D}$. It rewrites

---

[2] We recall that a term is either a variable or a constant in FO logic.

a query while ($i$) identifying its CQs with no answer on $\mathcal{D}$ using a summary $\mathcal{S}$ of it ((1) in Definition 4.4 below) and ($ii$) performing a bottom-up removal of the largest subqueries with no answer on $\mathcal{D}$ that these CQs are the cause of ((2) and (3) in Definition 4.4 below).

*Definition 4.4 (Summary-based optimization of a* $(\wedge, \vee)$*-CQ).* Let $q$ be a $(\wedge, \vee)$-CQ asked on a database $\mathcal{D}$ and $\mathcal{S}$ be a summary of $\mathcal{D}$ with the homomorphism $\sigma$. The *optimization of $q$ for $\mathcal{D}$ using $\mathcal{S}$*, i.e., denoted by $\Omega(q, \mathcal{S})$, is recursively defined as follows. Below, $\varnothing$ denotes the empty relation with appropriate arity.

The optimization of a CQ $q$ is:

$$\Omega(q, \mathcal{S}) = \begin{cases} \varnothing \text{ if } ans(q_\sigma, \mathcal{S}) = \emptyset \\ q \text{ otherwise} \end{cases} \quad (1)$$

where $q_\sigma$ is obtained from $q$ by replacing its constants by their images through $\sigma$.

The optimization of a conjunction of subqueries $\bigwedge_{i=1}^{n} q_i$ is:

$$\Omega(\bigwedge_{i=1}^{n} q_i, \mathcal{S}) = \begin{cases} \varnothing \text{ if } \exists i \in [1, n] \ \Omega(q_i, \mathcal{S}) = \varnothing \\ \bigwedge_{i=1}^{n} \Omega(q_i, \mathcal{S}) \text{ otherwise} \end{cases} \quad (2)$$

The optimization of a disjunction of subqueries $\bigvee_{i=1}^{n} q_i$ is:

$$\Omega(\bigvee_{i=1}^{n} q_i, \mathcal{S}) = \begin{cases} \varnothing \text{ if } \forall i \in [1, n] \ \Omega(q_i, \mathcal{S}) = \varnothing \\ \bigvee_{1 \leq i \leq n, \ \Omega(q_i, \mathcal{S}) \neq \varnothing} \Omega(q_i, \mathcal{S}) \text{ otherwise} \end{cases} \quad (3)$$

Above, the rewriting rule (1) follows from the soundness of identifying CQs with no answer using a database summary (Theorem 4.2), while the two other rewriting rules (2) and (3) follow from the semantics of the $\wedge$ and $\vee$ operators, respectively.

The next theorem establishes the two semantic relationships between a $(\wedge, \vee)$-CQ and its optimization, that correspond to items 1 and 2 in Problem 1. In particular, it states the correctness of summary-based optimization of a $(\wedge, \vee)$-CQ w.r.t. relational query evaluation.

THEOREM 4.5. *Let $\mathcal{D}$ be a database, $\mathcal{S}$ a summary of $\mathcal{D}$, and $q$ a $(\wedge, \vee)$-CQ. Then, $\Omega(q, \mathcal{S}) \subseteq q$ and $eval(q, \mathcal{D}) = eval(\Omega(q, \mathcal{S}), \mathcal{D})$.*

*Example 4.6 (Cont.).* The summary-based optimization of $q^{\text{UCQ}}$, $q^{\text{USCQ}}$ and $q^{\text{JUCQ}}$ for $\mathcal{D}$ using $\mathcal{S}$ corresponds to the following UCQ, USCQ and JUCQ, respectively. We also show in gray the subqueries that would have been additionally removed (with higher optimization time) if $\Omega$ had used $\mathcal{D}$ instead of $\mathcal{S}$.

$$\Omega(q^{\text{UCQ}}, \mathcal{S}) = (R(\mathsf{h}) \wedge ww(\mathsf{h}, x) \wedge sup(y, x)) \quad (1)$$
$$\vee (R(\mathsf{h}) \wedge ww(\mathsf{h}, x) \wedge PhD(x)) \quad (2)$$
$$\vee (R(\mathsf{h}) \wedge sup(\mathsf{h}, x)) \quad (3)$$

$$\Omega(q^{\text{USCQ}}, \mathcal{S}) = (R(\mathsf{h}))$$
$$\wedge (ww(\mathsf{h}, x) \vee sup(\mathsf{h}, x) \vee ww(x, \mathsf{h}))$$
$$\wedge (sup(y, x) \vee PhD(x))$$

$$\Omega(q^{\text{JUCQ}}, \mathcal{S}) = (R(\mathsf{h})) \wedge ((ww(\mathsf{h}, x) \wedge sup(y, x))$$
$$\vee (ww(\mathsf{h}, x) \wedge PhD(x))$$
$$\vee (sup(\mathsf{h}, x)))$$

For $\mathcal{L}_R \in \{\text{UCQ}, \text{USCQ}, \text{JUCQ}\}$, it can be easily checked that: $\Omega(q^{\mathcal{L}_R}, \mathcal{S}) \subseteq q^{\mathcal{L}_R}$ since $\Omega$ makes unions more specific by removing disjuncts, and $eval(\Omega(q^{\mathcal{L}_R}, \mathcal{S}), \mathcal{D}) = eval(q^{\mathcal{L}_R}, \mathcal{D})$ since both $q^{\mathcal{L}_R}$ and $\Omega(q^{\mathcal{L}_R}, \mathcal{S})$ model the CQ (3) that produces the sole answer w. ◇

## 4.2 Database summarization

The concrete database summaries that we use with our $\Omega$ optimization function are defined by adapting the quotient operation from graph theory [34] to the incomplete relational databases we consider. The quotient operation has been widely investigated in the literature for graph database summarization [18, 42]. It offers an elegant summarization technique by decoupling the summarization method, which basically fuses equivalent nodes, from the high-level specification of equivalent nodes, defined by an equivalence relation[3], e.g., bisimilarity [2]. Assuming we have an equivalence relation between database terms (the one we use will be discussed shortly), we define a *quotient database* as follows.

*Definition 4.7 (Quotient database).* Let $\mathcal{D}$ be a database, $\equiv$ be some equivalence relation between terms, and let $c_\equiv^1, \ldots, c_\equiv^k$ denote, by abuse of notation, both the equivalence classes of the terms in $\mathcal{D}$ w.r.t. $\equiv$ and the terms used to represent these equivalence classes.

The *quotient database of $\mathcal{D}$ w.r.t. $\equiv$* is the database $\mathcal{D}_\equiv$ such that:
• $R(c_\equiv^{\alpha_1}, \cdots, c_\equiv^{\alpha_n}) \in \mathcal{D}_\equiv$ iff there exists $R(term_1, \cdots, term_n) \in \mathcal{D}$ with $term_i \in c_\equiv^{\alpha_i}$ and $1 \leq \alpha_i \leq k$, for $1 \leq i \leq n$,
• the term $c_\equiv^j$ in $\mathcal{D}_\equiv$, for $1 \leq j \leq k$, is a variable if all the equivalent terms in $\mathcal{D}$ it represents according to $\equiv$ are variables, otherwise it is a constant.

The next proposition establishes that quotient databases can be used by the optimization function $\Omega$ to identify CQs with no answer on databases. It follows from the fact that in the above definition, $\equiv$ defines an implicit function that maps the terms in $\mathcal{D}$ to the terms in $\mathcal{D}_\equiv$, which turns out to be the homomorphism $\sigma$ in Definition 4.1: the first and second items in the above definition enforce respectively the conditions ($i$) and ($ii$) in Definition 4.1.

PROPOSITION 4.8. *Quotient databases are database summaries.*

We introduce the equivalence relation $\equiv_\Omega$ used to build our summaries, i.e., how database terms are fused into summary terms. Since ontology languages [6, 8, 15, 20] are centered on *concepts* modeled by unary relations, which are then interrelated using *relationships* modeled by n-ary relations, we adopt a summarization centered on the instances of concepts stored in a KB's database: all the terms that are instances of the same concept in the database are represented by a single term in the database summary (($i$) in Definition 4.9 below), and all the concepts with common instances in the database have the same single term that represents all their instances in the database summary (($ii$) in Definition 4.9 below). As we shall see in our experiments, $\equiv_\Omega$ achieves a good tradeoff between size reduction ($\geq$90%) and completeness of identifying CQs with no answer on the summarized databases (92% on average).

*Definition 4.9 ($\equiv_\Omega$ equivalence relation).* $\equiv_\Omega$ is the equivalence relation such that two terms $t_1$ and $t_2$ are equivalent within a database $\mathcal{D}$, denoted $t_1 \equiv_\Omega t_2$, iff ($i$) both $t_1$ and $t_2$ are terms of the same *unary relation*, i.e., concept, or ($ii$) there exists a term $t_3$ in $\mathcal{D}$ such that $t_1 \equiv_\Omega t_3$ and $t_2 \equiv_\Omega t_3$.

*Example 4.10 (Cont.).* The summary $\mathcal{S}$ in Example 4.3 is actually the quotient database of $\mathcal{D}$ w.r.t. $\equiv_\Omega$: it defines two equivalence classes, one for the researchers in $\mathcal{D}$, i.e., $\{\mathsf{f}, \mathsf{h}, u\}$, and one for the PhD students in $\mathcal{D}$, i.e., $\{\mathsf{w}, \mathsf{c}\}$; these two classes are represented in $\mathcal{S}$ by the constants $\mathsf{r}$ and $\mathsf{p}$, respectively. ◇

We discuss the need of summary maintenance in case of database updates in Section 6, as well as how our particular summaries (quotient databases w.r.t. $\equiv_\Omega$) can be efficiently updated.

---

[3]An equivalence relation is a reflexive, symmetric, and transitive binary relation.

## 5 EXPERIMENTAL EVALUATION

Our code, scripts, queries, data as well as the external resources and tools we used are available at or from: https://github.com/OptiRef/resources.git

**Setup.** For our KBs, we use the well-established *extended LUBM benchmark* a.k.a. $\text{LUBM}^\exists$ [43]. It is an adaptation of the Leight University benchmark a.k.a. LUBM [35] to the $\text{DL-lite}_{\mathcal{R}}$ description logic [17]. We chose this benchmark for two reasons. First, $\text{DL-lite}_{\mathcal{R}}$ is the most expressive KB language for which the reformulation of CQs into UCQ, USCQ and JUCQ reformulations has been studied. Second, this benchmark is widely-considered in the OMQA literature and provides opportunities to adapt many available queries to our needs. For the ontology $O$ of all our KBs, we used the default benchmark ontology $\text{LUBM}^\exists_{20}$. It is made of 449 positive rules over 163 relations: 128 unary relations, a.k.a. concepts, and 35 binary relations, a.k.a. roles. We used the EUGen (v0.1b) data generator provided with the benchmark to generate the databases of our KBs.

We used DB2 (v11.5.5), MySQL (v8.0.34) and PostgreSQL (v14.2) to store the generated databases and their summaries, which are commonly used in the OMQA literature. For space considerations, we report on the results obtained with the popular, open-source PostgreSQL RDBMS. We obtained comparable results with DB2 and MySQL. We adopted the data layout of [12] for the databases and summaries, which was found to be the most efficient for evaluating query reformulations on $\text{DL-lite}_{\mathcal{R}}$ KB's database. $\text{DL-lite}_{\mathcal{R}}$ uses unary relations for concepts, which are stored as unary tables, and binary relations for relationships, which are stored as binary tables. Moreover, all the values are dictionary-encoded into integers; the dictionary is stored as a binary table. Finally, for a database summary, the database-to-summary homomorphism $\sigma$, which maps the database terms to the summary terms, is stored as a binary table. For all the above-mentioned database, summary, dictionary and homomorphism tables, each unary table has an index on its unique attribute and each binary table has the two two-attributes indexes.

We used the Rapid (v0.93) [21], Compact (v1.0b6) [51] and GDL (v1.0) [12] FO-rewriting tools that respectively compute UCQ, USCQ and JUCQ reformulations of CQs w.r.t. $\text{DL-lite}_{\mathcal{R}}$ ontologies. They load and keep in memory the ontology w.r.t. which CQs are reformulated. While Compact and GDL are the only options to respectively compute USCQ and JUCQ query reformulations, there are other tools besides Rapid that can be used to compute UCQ query reformulations, e.g., Graal [7], Iqaros [52], Nyaya [53], Presto [47], Requiem [46], etc. Choosing Rapid instead of another tool does not affect our conclusions, as reformulation time is negligible w.r.t. both optimization and evaluation times: reformulation is performed w.r.t. the in-memory ontology, while optimization and evaluation is performed w.r.t. the on-disk data (summary and database).

Finally, to perform our experiments, we use a Ubuntu 20.04.2 Linux server with Intel Xeon 4215R 3.20GHz CPU, 128GB of RAM, and 7TB of fast HDD.

**Database summarization.** We generated five LUBM databases: LUBM1M, LUBM10M, LUBM50M, LUBM100M, LUBM150M. The name of a database indicates the number of stored facts in millions. Also, databases are created such that LUBM1M $\subseteq$ LUBM10M $\subseteq$ LUBM50M $\subseteq$ LUBM100M $\subseteq$ LUBM150M, where $\subseteq$ means set inclusion, so that query answering becomes harder as data grows.

| Database $\mathcal{D}$ | $|\mathcal{D}|$ | $|\mathcal{S}|$ | size red. (%) | sum. time (s) |
|---|---|---|---|---|
| LUBM1M | 1,187k | 93k | 92.12 | 15 |
| LUBM10M | 10,794k | 843k | 92.18 | 86 |
| LUBM50M | 53,328k | 4,160k | 92.20 | 308 |
| LUBM100M | 106,596k | 8,316k | 92.19 | 699 |
| LUBM150M | 159,899k | 12,474k | 92.19 | 1,100 |

**Table 3: Characteristics of the databases and summaries, database size reduction and summarization time for PostgreSQL**

We rely on the *union-find* data structure for disjoint sets [22] for database summarization, since equivalence classes of database terms w.r.t. $\equiv_\Omega$ are disjoint sets of equivalent terms w.r.t. $\equiv_\Omega$. This data structure supports two main operations, *union* and *find*, in optimal constant amortized time complexity [49, 50], i.e., time complexity is almost constant over a sequence of union or find operations. *Union* is used to state which values must be in a same set, and results in merging the sets these values belong to. *Find* returns the representative value of the set a given value belongs to.

We first compute the homomorphism $\sigma$ from the database $\mathcal{D}$ to the summary $\mathcal{S}$ (Definition 4.1) w.r.t. the $\equiv_\Omega$ equivalence relation (Definition 4.9). Given a union-find data structure for disjoint sets of integers, we use *union* to state that the (integer-encoded) terms stored in each unary relation in $\mathcal{D}$ must be in a same set, as these terms are equivalent w.r.t. $\equiv_\Omega$ (condition (*i*) in Definition 4.9). By definition of *union*, this ensures that if unary relations share some terms, in which case all the terms of these relations are equivalent w.r.t. $\equiv_\Omega$ (condition (*ii*) in Definition 4.9), then these terms end up in the same set. Finally, since *find* returns a representative term for the set of equivalent terms a given term belongs to, it models the homomorphism $\sigma$ from the database $\mathcal{D}$ to its summary $\mathcal{S}$ w.r.t. $\equiv_\Omega$. The computation of $\sigma$ is therefore linear in the size of the data: it needs a worst-case number of calls to *union* in the size of $\mathcal{D}$, each of which is performed in constant amortized time. Then, the summary $\mathcal{S}$ of the database $\mathcal{D}$ w.r.t. $\equiv_\Omega$ is computed as per Definition 4.7: every fact in $\mathcal{D}$ leads to a fact in $\mathcal{S}$ obtained by replacing each term by its image through $\sigma$, i.e., through *find*. The computation of $\mathcal{S}$ is therefore linear in the size of the data: it needs a worst-case number of calls to *find* in the size of $\mathcal{D}$ (one or two calls per fact), each of which is performed in constant amortized time.

Table 3 shows for each database $\mathcal{D}$ we generated: its size $|\mathcal{D}|$ and the size $|\mathcal{S}|$ of its summary $\mathcal{S}$, i.e., numbers of facts, the $\mathcal{D}$-to-$\mathcal{S}$ size reduction $(1 - |\mathcal{S}|/|\mathcal{D}|)$, and the summarization time with PostgreSQL (computation and storage of $\sigma$ and then of $\mathcal{S}$). We observe that $\equiv_\Omega$ achieves significant size reduction ($\geq$ 90%) and that summarization time scales linearly in the size of the data.

**OMQA performance.** We used ten CQs adapted from [12, 43] to obtain a variety of numbers of maximally-contained CQs w.r.t. $O$ that query reformulations model (recall Section 2) and of answers. The main characteristics of these CQs are shown in Table 4 (top).

For each database, we processed every query with 3 query answering strategies per $\mathcal{L}_R$ query reformulation languages used by FO-rewriting tools: $\mathcal{L}_R$ = UCQ for Rapid, $\mathcal{L}_R$ = USCQ for Compact and $\mathcal{L}_R$ = JUCQ for GDL. The first strategy, denoted by $\mathcal{L}_R$/REF, simply consists in computing the $\mathcal{L}_R$ query reformulation with the FO-rewriting tool and then evaluating it with PostgreSQL; this is how OMQA is performed via FO-rewriting, hence the state-of-the-art baseline. The second strategy, denoted by $\mathcal{L}_R$/DB, departs from

| Query | Query answering | | | | | | | | | |
|---|---|---|---|---|---|---|---|---|---|---|
| | QA0 | QA1 | QA2 | QA3 | QA4 | QA5 | QA6 | QA7 | QA8 | QA9 |
| #atoms | 8 | 5 | 5 | 6 | 6 | 8 | 8 | 8 | 6 | 8 |
| #contained CQs w.r.t. $O$ | 2,759 | 1,949 | 1,701 | 1,151 | 719 | 495 | 299 | 183 | 143 | 31 |
| #answers in $ans(q, \mathcal{K})$ (LUBM100M) | 23,946 | 0 | 347,527 | 720 | 69 | 0 | 2 | 858,259 | 12 | 0 |
| optimization ratio for UCQ/S | 81.92 | 100 | 99.88 | 83.80 | 80.2 | 80 | 52.53 | 78.89 | 66.91 | 77.42 |
| optimization ratio for USCQ/S | 100 | 100 | 100 | 100 | 100 | 100 | 100 | 100 | 100 | 100 |
| optimization ratio for JUCQ/S | 100 | 100 | 100 | 100 | 100 | 100 | 100 | 78.89 | 100 | 100 |

**Table 4: Characteristics of the queries (available at https://github.com/OptiRef/resources.git), $O = \text{LUBM}_{20}^{\exists}$ and $\mathcal{D} = \text{LUBM100M}$**

$\mathcal{L}_R$/REF by optimizing the query reformulation for the database $\mathcal{D}$ before evaluating it. For this strategy, our $\Omega$ function optimizes the query reformulation using the database $\mathcal{D}$. Finally, the third strategy, denoted by $\mathcal{L}_R$/S, is similar to $\mathcal{L}_R$/DB except that our $\Omega$ function optimizes the query reformulation for $\mathcal{D}$ using the summary $\mathcal{S}$ of $\mathcal{D}$.

Table 4 (bottom) shows the *optimization ratio* per query obtained with $\mathcal{L}_R$/S on LUBM100M, i.e., the percentage of CQs with no answers on LUBM100M that are identified and removed by $\Omega$ using LUBM100M's summary; the ratio is 0% with $\mathcal{L}_R$/REF and 100% with $\mathcal{L}_R$/DB. We observe that optimization ratios are high in general, 92% on average with 52.53% the lowest value ($QA6$), thus our summaries are effective to identify CQs with no answers. Similar results are obtained on LUBM1M, LUBM10M, LUBM50M, and LUBM150M.

Figure 2 shows the times we measured when we processed our queries with the above-mentioned strategies. For a given strategy, the measured time is defined as optimization time (for $\mathcal{L}_R$/DB and $\mathcal{L}_R$/S only) + evaluation time (recall item 3 in Problem 1). For space considerations, we report times for LUBM10M and the ten times larger LUBM100M. Generally, times gradually increase as the data size grows from 1M to 150M facts. Every reported time is an average over 5 "hot" query runs, i.e., the first "cold" query run is discarded.

$\mathcal{L}_R$/S ***versus the state-of-the-art baseline*** $\mathcal{L}_R$/REF**.** We observe that when query reformulations are optimized by $\Omega$ using $\mathcal{S}$:
• Query answering performance almost always improves for UCQs (UCQ/S for all the databases except for $QA6$), often significantly and up to more 3 orders of magnitude (e.g., UCQ/S for $QA1$ on LUBM10M and LUBM100M).
• Query answering performance frequently improves for JUCQs (in half of the cases overall), up to one order of magnitude (e.g., JUCQ/S for $QA8$ on LUBM100M), otherwise performance is marginally affected. We remark that when the performance visibly degrades (e.g., $QA9$ on LUBM100M) it is just in the order of a few tens of ms.
• Query answering performance is marginally affected for USCQs.

These observations are explained with the two following facts, and the optimization ratios obtained with our summaries (Table 4).
(1) Optimizing reformulations with $\Omega$ removes CQs with no answer from the top union in UCQs and from the unions on which the top join is performed in JUCQs; in USCQs, single-atom CQs are removed from unions on top of which joins are performed, on top of which the top union is performed.
(2) Removing CQs with no answer from a union improves its evaluation time (as it may take time for an RDBMS to find out that a CQ has no answer), while it does not change the size of its output hence the number of tuples to process after this union.

Therefore:
• Optimizing a UCQ reformulation with $\Omega$ speeds up its entire evaluation since $\Omega$ optimizes its top union. Also, because our summaries allow high optimization ratios for UCQ/S, query answering performance is significantly improved in general. We remark that performance degrades for $QA6$ because the optimization time does not amortize with a low optimization ratio (52.53% on LUBM100M).
• Optimizing a JUCQ reformulation with $\Omega$ speeds up the evaluation of its sub-UCQs but does not affect the evaluation time of the top join (as the same tuples must be joined). JUCQ reformulations are thus more difficult to optimize than UCQ ones. This is why query answering performance is "only" frequently improved (in half of the cases) and marginally affected otherwise, even with high optimization ratios for JUCQ/S (> 78% on LUBM100M).
• Optimizing a USCQ reformulation with $\Omega$ only removes atomic CQs from its inner unions while it does not take time for an RDBMS to figure out that these atomic CQs are empty. The optimization thus marginally affects the evaluation time of these inner unions, and the evaluation time of the subsequent joins and top union is not affected. USCQ reformulations are thus more difficult to optimize than UCQ and JUCQ ones. This is why query answering performance is marginally affected in general, even with maximal optimization ratios for USCQ/S (100% on LUBM100M).

$\mathcal{L}_R$/DB ***versus*** $\mathcal{L}_R$/REF ***and*** $\mathcal{L}_R$/S**.** We observe that when $\Omega$ optimizes query reformulations using $\mathcal{D}$ instead of $\mathcal{S}$, query answering performance may improve or degrade:
• Query answering performance is marginally to significantly better with UCQ/DB than with the baseline UCQ/REF, although the performance with UCQ/DB is generally worse than with UCQ/S (except for $QA6$ that has a low optimization ratio of 52.53%).
• Query answering performance with JUCQ/DB is always worse than with the baseline JUCQ/REF and almost always worse than with JUCQ/S (except for $QA7$ on LUBM10M and for $QA7$ and $QA8$ on LUBM100M).
• Query answering performance with USCQ/DB is always worse than with the baseline USCQ/REF and with USCQ/S.

These observations are explained by the extra-time spent by $\mathcal{L}_R$/DB w.r.t. $\mathcal{L}_R$/S in completely optimizing a query reformulation using the database (recall that optimization ratios are of 100% for $\mathcal{L}_R$/DB): optimization time with $\mathcal{L}_R$/DB is in general significantly higher than with $\mathcal{L}_R$/S, because a database is much larger than its summary, while at the same time $\mathcal{L}_R$/DB provides a moderate gain in optimization ratios because they are already very high with $\mathcal{L}_R$/S in general. This is why $\mathcal{L}_R$/DB performs worse than $\mathcal{L}_R$/S

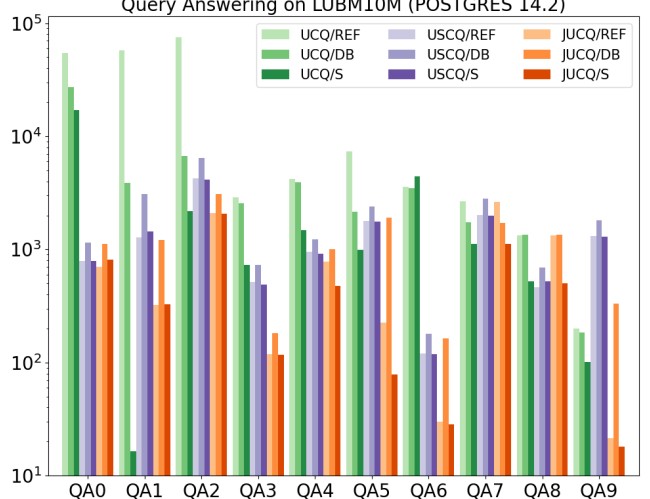
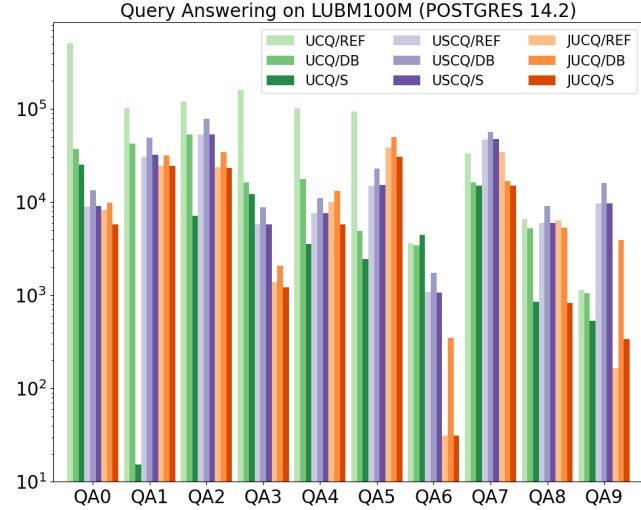

**Figure 2: Query answering times (ms, logscale) with PostgreSQL on LUBM10M (left) and LUBM100M (right)**

overall, and worse than $\mathcal{L}_R$/REF when optimization time is higher than the time saved when the optimized reformulation is evaluated.
**Conclusion.** Our experiments show that our summaries can be fast to compute (linear in data size), small (<10% of data size) and effective to identify CQs with no answer on a database (92% on average). They also show that, when our $\Omega$ optimization function uses summaries, OMQA time performance can be significantly improved (item 3 in Problem 1) for UCQ reformulations in general and frequently for JUCQ reformulations, while performance is marginally affected for USCQ ones.

## 6 RELATED WORK AND CONCLUSION

We devised a novel optimization framework for OMQA via FO-rewriting. Our framework is complementary to, and capitalizes on, the optimizations that have been proposed so far in the literature, e.g., [11, 12, 21, 33, 38, 51]. These optimizations are both ontology-dependent and data-independent. They exploit the ontology's rules to find equivalent query reformulations that can be evaluated faster on relational databases, similarly to semantic query optimization that exploits the rules of deductive databases, e.g., [19]. The novelty of our framework is to add a complementary data-dependent optimization step to query reformulations produced by state-of-the-art FO-rewriting tools, e.g., [7, 11, 12, 21, 46, 47, 51–53]. This framework is general enough to apply to a variety of FO-rewriting settings, in particular those in Table 2, and it guarantees the correctness of OMQA on the queried KBs. For the FO-rewriting settings in which it was evaluated, it significantly improves OMQA time performance for the widely-adopted UCQ query reformulations, e.g., [10, 17, 21, 30–33, 38, 45–47, 52], and for the JUCQ ones of [11, 12]. Finally, an originality of our framework is that it builds on the $\Omega$ optimization function that rewrites a query reformulation into a simpler contained one, by pruning away subqueries that are useless to its evaluation on a given database. Notably, useless subqueries are identified rapidly by using database summaries, which we devised for this particular purpose by adapting the quotient operation [34] to databases.

**Alternative summaries.** Database summaries, in particular those based on the quotient operation, have been mainly investigated for graph databases, e.g., [18, 42], and description logic databases a.k.a. ABoxes [23–25, 27] for the purpose of data exploration and of data management optimization (consistency checking and query answering). To the best of our knowledge, summaries have not been used for the optimization of OMQA via FO-rewriting. We adapted the quotient operation to relational databases and we defined the new equivalence relation $\equiv_\Omega$ for the special task of sound and fast identification of CQs with no answer on a database. $\equiv_\Omega$ departs from prior equivalence relations by being based on the instances of concepts that KB's databases describe with n-ary relationships between them, and not on bisimulation [36], e.g., [26, 44], or cooccurrence of relationships [28, 29]. A perspective is to study alternative database summaries for our framework, which could improve further OMQA time performance: summaries could be obtained either via the quotient operation and other equivalence relations than $\equiv_\Omega$, or with other procedures than the quotient operation.

**Summary maintenance upon database updates.** Although the computation of our summaries is linear in the size of databases, the summarization times we reported in our experiments show that it would be prohibitive to redo full summarization upon updates. We therefore rely on incremental summary maintenance. We remark that the need for incremental maintenance is shared with the two other OMQA techniques, materialization, e.g., [30, 41] and combined approach, e.g., [39, 40, 43], though we need to maintain a summary that is a small and simple homomorphic approximation of the KB's database, while materialization and combined approach need to maintain a large and complex (approximation of a) chase of the KB's database, i.e., database plus entailed facts. By definition of a summary built with $\equiv_\Omega$, in the worst case, an insertion fuses two equivalence classes and a deletion splits an equivalence class into several ones. Maintenance rewrites the affected summary facts, i.e., in which some term moves from an equivalence class to another, based on the updated homomorphism $\sigma$ modeled with a union-find data structure (recall Section 5) that also supports the delete operation in optimal constant amortized time complexity [4].

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

# APPENDIX

PROOF OF THEOREM 4.2. We prove the theorem by showing that its contrapositive holds, i.e., *if q has some answer on $\mathcal{D}$, then $q_\sigma$ has some answer on $\mathcal{S}$.* If q has an answer on $\mathcal{D}$, then there exists a homomorphism $h$ from $q$ to $\mathcal{D}$ such that $h(q) \subseteq \mathcal{D}$ where every free variable is mapped to a constant, every existential variable is mapped to a constant or variable, and every constant is mapped to itself. Moreover, the composition $\sigma \circ h$ is a homomorphism from $q$ to $\mathcal{S}$ such that $\sigma \circ h(q) \subseteq \mathcal{S}$ where, by definition of a database summary, every free variable is mapped to a constant, every existential variable is mapped to a constant or variable, and every constant is mapped to its image through $\sigma$. Let us now build a homomorphism $g$ from $q_\sigma$ to $\mathcal{S}$ such that $g(q_\sigma) = \sigma \circ h(q) \subseteq \mathcal{S}$: it suffices that $g$ maps every variable exactly as $\sigma \circ h$ does, while it maps every constant to itself (constants have already been replaced by their image through $\sigma$ in $q_\sigma$). Since defined this way $g$ maps free variables to constants, $q_\sigma$ has an answer on $\mathcal{S}$. □

Proof of Theorem 4.5. Let us first prove $\Omega(q, \mathcal{S}) \subseteq q$. We prove this by induction on the depth $d$ of $q$ defined as the maximal nesting of the $\wedge$ and $\vee$ operators on top of CQs, with the *induction hypothesis* that $\Omega$ performs rewritings (rules (1), (2) and (3) in Definition 4.4) that are contained in the rewritten query. *Base case*, $d = 0$: rule (1) rewrites $q$ either by (second case) itself or by (first case) $\varnothing$, and clearly, $q$ is contained in itself and $\varnothing$ is contained in $q$. *Induction step*, $d > 0$: rule (2) rewrites a conjunction either by (second case) a contained one (induction) or by (first case) $\varnothing$ that is by definition contained in the rewritten conjunction; rule (3) rewrites a disjunction either by (second case) a contained one (induction), or by $\varnothing$ (first case) that is by definition contained in the rewritten disjunction.

Let us now prove that $eval(q, \mathcal{D}) = eval(\Omega(q, \mathcal{S}), \mathcal{D})$. Again, we prove this by induction on the depth $d$ of $q$ defined as the maximal nesting of $\wedge$ and $\vee$ operators on top of CQs, with the *induction hypothesis* that $\Omega$ performs rewritings (rules (1), (2) and (3) in Definition 4.4) that are equivalent w.r.t. the database $\mathcal{D}$. *Base case*, $d = 0$: rule (1) rewrites $q$ either by (second case) itself or by (first case) $\varnothing$ if $q$ has no answer on $\mathcal{S}$, hence on $\mathcal{D}$ according to Theorem 4.2, i.e., $q$ is equivalent to $\varnothing$ on $\mathcal{D}$. *Induction step*, $d > 0$: rule (2) rewrites a conjunction either by (second case) an equivalent one (induction) or by (first case) $\varnothing$ if a $q_i$ subquery has no answer on $\mathcal{D}$ (induction), hence the conjunction is equivalent to $\varnothing$ on $\mathcal{D}$; rule (3) rewrites a disjunction either by (second case) an equivalent one (induction), or by $\varnothing$ (first case) if all its subqueries have no answer on $\mathcal{D}$, hence the disjunction is equivalent to $\varnothing$ on $\mathcal{D}$. □

