# OpenReview forum: "Query Optimization for Ontology-Mediated Query Answering"
_ACM.org/TheWebConf/2024/Conference — TheWebConf24 Oral_

### Official Review · Reviewer_oeDs · 2023-11-17

**Novelty:** 6
**Technical Quality:** 6

**Review:**

# Overview:
The paper studies the classical ontology based query answering but with a twist: while the traditional approach was to rewrite the ontology part into a huge query that works over any suitable set of base data, introducing potentially huge cost in terms of query evaluation, in the current paper the authors prune this rewriting so that it keeps only the part of the query relevant for the database we have at hand. An additional optimization which does this pruning based on the summary of the base data is also described and implemented. The proposed solution is quite simple, sound, and seems to work well in practice.

# Strengths:
- The setting/studies problem is very interesting and relevant. Approach which simply pushes all the logic/inference into the query rewriting had been the norm until now, and the optimization that is proposed here (which has to obtain this rewriting as well) is quite natural for this setting. Description of the studied problem in Problem 1 is spot on in that sense.
- Overall, the writing is quite clear, the proofs are correct, and it is easy to understand the proposed solution.
- The experimental data is available in a github repository.
- The experimental setup is decent and it nicely showcases for which class of queries the proposed solution works well, and where it runs into roadblocks.

# Weaknesses:
- The zime complexity aspect of Problem 1 was never fully tackled. One could argue that the experimental evaluation does the trick here, but some theoretical guarantees would also be nice I guess. This is particularly relevant since the q^\fancyO rewriting can already be quite big, and decomposing it further might cause some more blowups (I am not sure if I am off here, but distribution ors over ands can cause exponential rise in the number of terms generally). Perhaps some discussion on this issue is warranted.
- The writing, while generally excellent, is lacking several details in places. For someone reviewing query containment papers for the past twenty years this might not be an issue, but a non expert user might be lost at times. I provide concrete suggestions for tightening the writing below.
- Dwelling a bit deeper into different data summaries might be interesting for the experimental evaluation, but I understand that the space is very limited.
.

# Recommendation:
Overall, I would be more than happy to provide my support for this paper. While the proposed solution is quite simple and natural, I think it fills a nice niche in the area of ontology mediated query answering. Pushing this further, perhaps the real contribution is the setup for this problem to be stated. The solution itself is the almost trivial, but getting there might not be and I would like to fully acknowledge this.

# Post Rebuttal:
I would like to thank the authors for their very detailed responses which indeed clarified my doubts. I will stick with my original recommendation and am suggesting the paper to be accepted.

# Some comments on writing:
- On page 2, when defining query answers it should be either said that you consider logical implication, or define what certain answers are (providing references as well).
- On page 3, relational evaluation of q^\fancyO is not clear since the database can have variables. Please clarify.
- At the beginning of section 4.1. I am getting a bit lost with the nomenclature. Most notably as to why ww(h,x) is derived. A similar confusion arises when reading Theorem 4.5. I guess that the issue is the fact that we lose the context of the presentation. Perhaps it would be worth stressing again here that we are given q^\fancyO and we run from there. This q^\fancyO already derived all the facts it needed, so we just look at the logical structure of the formulas we are processing.
- On page 5, when defining the quotient \sigma, why not just spell out the formula, e.g. \sigma(t)=c^i_\equiv , where t\in c^i_\equiv.
- The use of "e.g." in the Intro is off.

**Questions:**

- Are there any particular reasons why UCQ optimizations have the lowest optimization ratio in Table 4? From the time results I would expect the opposite behavior, but perhaps I misunderstood the optimization ratio metric.
- What is the size of the q^\fancyO rewritings obtained for LUBM ontology?
- As a curiosity, since a relatively large disk is used, are any space bottlenecks for the implementation? I know these would just boil down to postgres, but possibly the rewriting of tested queries push it quite far?
- Did the authors try to scale the experiment to e.g. 1B facts? Does the solution scale to this degree?

**Ethics Review Description:**

No issues detected

**Reviewer Confidence:**

4: The reviewer is certain that the evaluation is correct and very familiar with the relevant literature

**Scope:**

3: The work is somewhat relevant to the Web and to the track, and is of narrow interest to a sub-community

---

### Official Review · Reviewer_uAN2 · 2023-11-21

**Novelty:** 5
**Technical Quality:** 5

**Review:**

This paper deals with the problem of query optimization when ontological knowledge needs to be taken into account. There are several approaches to this problem, the most common one being to rewrite the query in order to take into account ontological knowledge. This is a generic method, that does not consider the current contents of the database, only the ontological knowledge, and, thus, it is applicable to all possible databases.

The authors' work takes this approach one step further and considers the underlying database as well during query rewriting. The way this is done is by taking the generic query reformulation (rewriting) above, and removing conjunctive query clauses (CQs) that definitely have no answer in the specific database. The identification of the CQs with no answer can be done by querying the database itself, or a properly generated summary of the database (the latter is more efficient, of course). This is experimentally shown (using the LUBM benchmark) to improve query evaluation performance, in the general case.

The paper is well-written, albeit a bit verbose at places. The considered problem is an important one. The method is rather simple and not highly sophisticated (especially the summary-generation process; see below) but seems sound and effective for its stated purposes.

My main comments are associated with the summary-generation process, which I found a bit simplistic. It is clear that the effectiveness of the authors' method depends on the form of the summary. The chosen summary-generation process essentially collapses all instances of the same class into one. This has the side-effect that if there are instances belonging to multiple classes then this will cause all instances of all involved classes to collapse as well. And this can have cascading effects, essentially collapsing large parts of the database.

To paraphrase the authors' example, suppose that we have two more classes, male and female. Since we have male PhD students (and supervisors) and female PhD students (and supervisors), we will end up with a summary having a single instance, essentially collapsing the members of two classes that are supposed to be disjoint (PhD students and supervisors). The fact that disjoint classes can collapse will cause many false negatives in the process of identifying CQs with no answers.

Note that this is a common pattern in rich ontologies, where each instance may be classified against multiple different characteristics (e.g., profession, gender, nationality, ...) which are orthogonal to each other and will cause "collapses" in the above sense. This will significantly hinder the effectiveness of the method, leading to a lot of false negatives as regards the identification of empty CQs.

Despite the fact that the experimental evaluation showed good results under this summary-generation algorithm, I'm still concerned about its effectiveness in the general case. Note that the experiments only considered one single dataset (LUBM), and LUBM, as far as I remember, does not have Male/Female classes or other such major "orthogonal" classifications.

Also, are implicit instantiations considered during the summary generation? I suppose (and hope) that this is not the case, but this is not clarified in the paper. If so, the existence of a general class (e.g., similar to owl:Thing or rdf:Resource) will ruin any chance of identifying empty CQs. I'm pretty sure the answer is "no", otherwise the mere existence of the class "Person" in LUBM would cause the approach to fail.

On Section 5: the authors provide only some of the figures of their experimental evaluation due to space considerations. It would be good to have the remaining ones in the appendix (like the theorem proofs).

Typo:
- "up to more 3 orders"

I wish to thank the authors for their comments, clarifications and acknowledging the observation about efficiency.
It is understood that the modelling pattern I mentioned is not ubiquitous. However, it exists, and thus limits the applicability of the authors' approach. I understand and agree that the problem could be resolved with an alternative summarization method. I believe that considering the effect of different summarization methods could be a nice addition to a future paper.

**Questions:**

See main review, in particular the comments about the summary-generation process.

The authors are asked to comment on the comments stated above.

**Ethics Review Description:**

i selected NO above and yet  description is needed. Please fix the bug.

**Reviewer Confidence:**

3: The reviewer is confident but not certain that the evaluation is correct

**Scope:**

4: The work is relevant to the Web and to the track, and is of broad interest to the community

---

### Official Review · Reviewer_3U1r · 2023-11-24

**Novelty:** 6
**Technical Quality:** 6

**Review:**

The paper considers a general scenario of ontology-mediated query answering by query rewriting.

The details vary, but the general idea is to rewrite queries in the presence of an ontology such that the answers to the rewritten query, without any more reasoning, are the same as the certain answers to the original query taking into account the ontology.

It is central to this approach that the rewritten query is independent of the actual data.  As a consequence however, it will often have poor performance on specific datasets because of parts of the query being redundant.

This work suggests an approach where the rewritten query is further optimised using the dataset, while still preserving the query answers.  To avoid the inefficiencies of considering the whole database for this purpose, the method works on a "summary" of the database.  Depending on the summary used, the optimisation will be more or less effective at removing redundant parts of the query.

I find this to be valuable work in general, and novel as far as I can tell.  I have a few questions below that I would like to have answers.

**Questions:**

1) the statement of Problem 1 contains requirement (1), followed by an explanation that other optimisations, that are not subqueries of the original, are not of interest.  I ask why? If the algorithm can guarantee that rewriting "Where does TheWebConf 2024 take place" to "Where does Petra live" is sound FOR THIS DATASET, and also that doing this optimisation and evaluating the resulting query will take less time than evaluating the original query, what is wrong with that?

It seems to me that requirement (1) is a property of your solution and not something necessarily required for solving the actual problem.

2) The use of the computation time τ(.) in a formal definition is problematic. The computation time is not well defined and  Can this be replaced by a reasonable formally defined metric on queries?

3) The problem of irrelevant disjuncts in query rewritings has been studied previously.  E.g., the following paper considers the problem in the case where the dataset is described not by a summary, but by certain kinds of constraints:
OBDA Constraints for Effective Query Answering, https://link.springer.com/chapter/10.1007/978-3-319-42019-6_18

In general, I would have liked to see more comparison of this summary-based approach to that of optimising the rewritten query based on database constraints, SHACL shapes, etc.

4) The approach is presented as an integral part of an OMQA approach. It seems to me that the paper could be simply about the optimisation of queries using dataset summaries. Possibly the types of redundancies the presented approach is effective for are particularly prevalent in the result of FO rewriting?

**Reviewer Confidence:**

3: The reviewer is confident but not certain that the evaluation is correct

**Scope:**

3: The work is somewhat relevant to the Web and to the track, and is of narrow interest to a sub-community

---

### Official Review · Reviewer_b2Sf · 2023-11-25

**Novelty:** 4
**Technical Quality:** 4

**Review:**

The paper proposes optimization techniques for Ontology-Mediated Query Answering.  The proposed approach makes use of the FO-writing to fist formulate the required query, which is then further optimized  by computing  simpler (contained) queries with same
answers that can be evaluated faster by RDBMSs. The optimization makes use of a
a KB’s database summary. The evaluation is based on LUBM benchmark and the results are promising.

In general, the topic of the paper is relevant to the conference topics. Unfortunately, I found the paper hard to follow in general. May be some running examples could made it easy to follow.

**Questions:**

Q1. The evaluation results are based on only 9 queries. Do you really think that these queries are sufficient to draw solid conclusions ?

Q2. How the selected techniques for comparison with the proposed approach are state of the art?

Q3.  I am little curious if ontology-mediated query answering is used in practice ? some real-world usage might be interesting to discuss.

**Ethics Review Description:**

Nothing

**Reviewer Confidence:**

2: The reviewer is willing to defend the evaluation, but it is likely that the reviewer did not understand parts of the paper

**Scope:**

4: The work is relevant to the Web and to the track, and is of broad interest to the community

---

### Official Review · Reviewer_KbRj · 2023-11-26

**Novelty:** 3
**Technical Quality:** 3

**Review:**

The submission studies the problem of ontology-mediated query answering. The focus is on opmimizing query rewritings using a summary of the database. Some experiments are conducted and show that the technique works well for rewritings in the shape of unions of conjunctive queries (UCQs) and in half of the cases for rewritings in the shape of joins of UCQs (JUCQs). UCQs are prone to becoming very large in practice, but the fact that the optimization works there is hardly surprising - most of the combinations of atoms do not co-occur in the test database (which is what the submission actually demonstrates). JUCQs in general actually include UCQs, but the authors focus on one particular way of generating them (and this is not very well explained in the submission), which is why the presented results are somewhat surprising. To sum up - the experimental evaluation is not particularly extensive and a bit difficult to draw any conclusions from.

The list "datalog\pm and existential rules, description logic and OWL, or RDF/S" does not make much sense: the "and" seem to join alternative names for the same languages; also, RDF/S normally denotes "RDF or RDFS" - as there is not much FO-rewriting in plain RDF, it should really be RDFS, but then it's subsumed by OWL. More seriously though, this list tries to cover too much in too few words (additional explanations are needed). And it's used twice in the text - in the abstract and then again on p 1, without any elaboration.

The note that "non-recursive datalog programs ... unfold into UCQs" (p 1) is irrelevant and actually misleading - JUCQs and USCQs also unfold into UCQs, but it does not characterize them. The point is that non-recursive datalog not only generalizes JUCQs and USCQs, but also allows shared subqueries without any duplication.

The difference between minimal and compact reformulations is unclear (without reading the cited papers)? Normally, minimal would be compact too...

The authors claim novelty and originality of the optimization framework in Introduction and then re-iterate it in Conclusions. However, if we look at Example 4 in [11], then we'll also see an example of a data-dependent optimization: (1), (4) and (8) are derived from (0) by instantiating variable y with the three class names --  this could only work if we know that these are the only classes in the triple store (in other words, we have a summary of the triple store, which tells us which classes are empty and which are not - what if someone adds a triple of the form IRI rdf:type :CrimeNovel?). Such optimizations are quite natural and implemented, for example, in Ontop:

[*] R. Kontchakov, M. Rezk, M. Rodriguez-Muro, G. Xiao, M. Zakharyaschev: Answering SPARQL Queries over Databases under OWL 2 QL Entailment Regime. ISWC (1) 2014: 552-567

So, perhaps, more credit needs to be given to the authors of [11] and [*1].  Also, there is a considerable amount of work on using integrity constraints on the data in opmimizing query answering: e.g.,

[*2] J. Mora, R. Rosati, O. Corcho: kyrie2: Query Rewriting under Extensional Constraints in ELHIO. ISWC (1) 2014: 568-583

Integrity constraints (or ABox dependencies) can also be viewed as "summaries" of the data (they are certainly data-dependent).

The explanation "a simpler contained one, i.e., a simpler more specific one, with the same answers on a fixed database" is incomplete and misleading: first, the notion of query containment is not defined (and "a contained one" is actually quite an awkward way of bringing query containment into the sentence; second, "a more specific one" does not add any clarity to it; third, "the same answers" does not correspond to query containment (which only guarantees the subset relation between answers).

In the definition of FO queries, the authors could be more specific - the proposed technique does not work with queries containing negation (even in the contexts where it can be eliminated, e.g., \neg (\neg A \lor \neg B)), so, perhaps, it would make sense to concentrate on existential positive queries from the very beginning.

The claim on p 3 that "datalog-nr reformulations must be unfolded into UCQs reformulations... to be evaluated by RDBMSs" is not factually correct - some DB engines support enough of the Common Table Expression (CTEs) to deal with non-recursive datalog. It may be not the most efficient support, but saying "must be unfolded" is clearly incorrect.

The authors list 4 papers for "the worst-case number of CQs that are maximally-contained in a CQ ... is exponential in the size of the CQ" - but really, the result is a simple observation made in [17] - the rest of the references are not needed. Also, what is "a lightweight RDFS"? What is datalog\pm0?

The paragraph after Problem 1 does not clarify anything at all. Why is this a bad "optimization"? It delivers the required results, does it not?

The first item in the definition of quotient database (page 5) is unreadable.

In Definition 4.9, is it not easier to say "the minimal equivalence relation containing all (t1,t2) such that both t1 and t2 are terms of the same unary relation"? It also needs to be made clearer that the same unary relation in D. But what is a unary relation in D? "Concept" does not clarify this, as A \sqcap B and \exists R.C are also concepts in DLs. Is the vocabulary of classes and properties assumed to be fixed in advance? Is this "unary relation" then a class name (from a fixed vocabulary)? Fixing vocabulary in advance is not very typical in RDF (and in DLs and logic in general, it is quite common to assume a countably infinite vocabulary).

In Section 5, the "etc." in the list of rewriters is missing some names, e.g., Clipper.

The Conclusions section raises an important issue of computing and maintaining the summary. And this is where the usefulness of the proposed technique becomes less clear. First, in the materialization-based approaches, the "large and complex chase" can be stored in a compact way but it makes the query rewriting and answering steps very easy (in fact, almost trivial). The main drawback here is actually the need to have "write access" to the data - incrementally or not, the data needs to be extended. The rewriting approaches, on the other hand, have a penalty of expensive query rewriting but are applicable where the data cannot be changed or extended. The proposed approach seems to take the worst of the two - it does require some sort of "write access" to data and it also requires a potentially expensive query rewriting step. The argument with incremental updates does not really improve the situation - of course, we can imagine some sort of triggers that incrementally update the summary stored in some temporary tables, but that is still a sort of access that is often unavailable. If it is available, then why not go whole hog and materialize the chase?

Typos:

line 23: same -> the same

line 204: something is missing after the conjunctions over rules and facts or is it meant to be \bigwedge O and \bigwedge D?

Footnote 1 is unnecessary - it's really basic stuff (but looks too technical for the level of the material).

In DL-Lite_R, the L in Lite should be capital.

Footnote 2 is poor as it introduces a notion.

In line 498, the comma after "Then" should be removed

It's quite unusual to have a Conclusions paragraph in individual section (such as Section 5).

**Questions:**

Could the authors clarify how the submission fits the Call for Papers? RDF and RDFS are briefly mentioned in the text, but are not really an essential component - the technique (as the authors write) applies to existential rules, and this conference, perhaps, is not the best place for a paper on existential rules.

Also, any comment on why this approach is better than materialization (given that incremental updates of the summary would cost more or less the same)?

**Reviewer Confidence:**

4: The reviewer is certain that the evaluation is correct and very familiar with the relevant literature

**Scope:**

2: The connection to the Web is incidental, e.g., use of Web data or API

---

### Decision · Program_Chairs · 2024-01-22

**Decision:**

Accept (Oral)

**Comment:**

* (+) There are no concerns with scope
 * (+) Novelty and technical quality scores are largely uniform.
 An accepted paper should include promised changes/improvements/clarifications from the discussion period